# Adjuvant Icotinib in EGFR-mutated stage IB non-small cell lung cancer with high-risk factors: A retrospective case series

Mengzhi Cheng[1,2], Jianbin Zhang[1,2*], Lili Jin[3,4], Caihua Yu[1,2], Zhonghai Xie[1,2], Dong Li[1,2], Qinhua Gu[1,2], Qibin Shen[1,2]

1 Department of Thoracic Surgery, Huzhou Central Hospital, Fifth School of Clinical Medicine of Zhejiang Chinese Medical University, Huzhou, Zhejiang, China, 2 Department of Thoracic Surgery, Huzhou Central Hospital, Affiliated Central Hospital of Huzhou University, Huzhou, Zhejiang, China, 3 Department of Central Laboratory, Huzhou Central Hospital, Fifth School of Clinical Medicine of Zhejiang Chinese Medical University, Huzhou, Zhejiang, China, 4 Department of Central Laboratory, Huzhou Central Hospital, Affiliated Central Hospital of Huzhou University, Huzhou, Zhejiang, China

* fishing11@163.com

## Abstract

Primary results of the CORIN trial indicated that, compared with chemotherapy, icotinib significantly improved 3-year disease-free survival (DFS) in patients with Epidermal Growth Factor Receptor (EGFR)-mutated stage IB non-small cell lung cancer (NSCLC). However, evidence regarding the outcomes of adjuvant icotinib in patients with high-risk factors remains limited. This retrospective study evaluated the efficacy and safety of adjuvant icotinib in patients with EGFR-mutated high-risk stage IB NSCLC. We enrolled 37 patients with completely resected EGFR-mutated high-risk stage IB NSCLC. The median follow-up time was 31 months, and the 3-year DFS rate was 91.4%. Two patients experienced disease recurrence and were successfully switched to osimertinib upon identification of an EGFR (T790M) mutation. Although overall survival (OS) and central nervous system (CNS)-DFS data were not mature, no deaths or central nervous system metastases were observed by the end of follow-up. 29 (78.4%) patients experienced grade 1–2 adverse events (AEs), no grade 3 or higher AEs occurred. This study suggests a potential DFS benefit and well-tolerated profile of adjuvant icotinib in patients with EGFR-mutated high-risk stage IB NSCLC. However, longer-term follow-up is necessary to assess the long-term outcomes.

## Introduction

Lung cancer remains one of the most prevalent malignancies worldwide and is associated with high morbidity and mortality [1]. The two primary types of lung cancer are non-small cell lung cancer (NSCLC) and small cell lung cancer (SCLC). Data from the World Health Organization (WHO) in 2022 indicated that NSCLC accounts for

**Data availability statement:** All relevant data are within the manuscript and its Supporting Information files.

**Funding:** This research was supported by the Joint Funds of the Zhejiang Provincial Natural Science Foundation of China under Grant (No. LBY22H200006).

**Competing interests:** The authors have declared that no competing interests exist.

**Abbreviations:** EGFR: Epidermal Growth Factor Receptor; NSCLC: non-small cell lung cancer; DFS: disease-free survival; OS: overall survival; CNS-DFS: central nervous system-related disease-free survival; AEs: adverse events; NCCN: National Comprehensive Cancer Network; TKIs: tyrosine kinase inhibitors; LVI: lymphovascular invasion; STAS: spread through air spaces; VPI: visceral pleural invasion; IQR:Interquartile Range; CTR: consolidation tumor ratio; MRI: cranial magnetic resonance imaging; CT: Chest computed tomography; PET-CT: Positron Emission Tomography–Computed Tomography.

approximately 81.3% of all lung cancer cases [2]. Although a comprehensive treatment strategy centered on surgery can improve survival in patients with early-stage NSCLC, long-term outcomes remain unsatisfactory. The 5-year overall survival (OS) rate ranges from 67% at stage IB to 39% at stage IIIA disease [3]. Consequently, identifying strategies to further improve the therapeutic efficacy of early-stage NSCLC remains a important topic in the field of lung cancer.

Epidermal growth factor receptor (EGFR) gene is one of the most frequently mutated driver genes in NSCLC [4]. Due to their remarkable efficacy and safety profiles, EGFR tyrosine kinase inhibitors (TKIs) have been established for the first-line therapy in advanced EGFR-mutated NSCLC [5]. Moreover, numerous studies have also confirmed the superior efficacy of EGFR-TKIs over platinum-based chemotherapy in the adjuvant setting for radically resected EGFR-mutated early-stage NSCLC. According to findings of the ADAURA trial [6], osimertinib can significantly reduce the risk of recurrence and death in patients with EGFR-mutated stage IB to IIIA NSCLC. The ICTAN trial [7] demonstrated that adjuvant icotinib for 6 or 12 months after chemotherapy significantly improved survival outcomes in patients with stage II–IIIA disease. Furthermore, the CORIN trial [8] specifically reported that icotinib significantly extends the 3-year disease-free survival (DFS) in patients with completely resected EGFR-mutated stage IB NSCLC. However, evidence on the benefits of adjuvant EGFR-TKIs for stage IB NSCLC with high-risk factors remains limited. Therefore, we conducted this retrospective study to assess the efficacy of icotinib in this specific patient population.

## Methods

### Study design and eligibility criteria

This was a single-center, retrospective case-series study. From January 1, 2018 to September 30, 2023, 197 NSCLC patients who received postoperative adjuvant icotinib were initially enrolled. The data cutoff date was March 1, 2025. The inclusion criteria were as follows: 1) NSCLC patients who underwent completely surgical resection via video-assisted thoracoscopic surgery (VATS) or thoracotomy were eligible. 2) Pathological diagnosis as stage IB NSCLC according to the 8th edition of the UICC/AJCC TNM staging system. 3) Presence of one of the following high-risk factors: lymphovascular invasion (LVI), spread through air spaces (STAS), visceral pleural invasion (VPI), or poor cellular differentiation. 4) EGFR exon 19 deletion or exon 21 L858R mutations, initially identified in tumor tissue by a Polymerase Chain Reaction-Amplification Refractory Mutation System(PCR-ARMS) assay and confirmed using a 10-gene panel. The exclusion criteria were as follows: 1) A history of lung cancer or multiple primary lung cancers. 2) Dose reduction, interruption, or discontinuation for non-medical reasons (Fig 1).

This study was approved by the Research Ethics Committee of Huzhou Central Hospital, Affiliated Central Hospital of Huzhou University (NO. 202112034−01). Written informed consent was obtained from all participants before treatment.

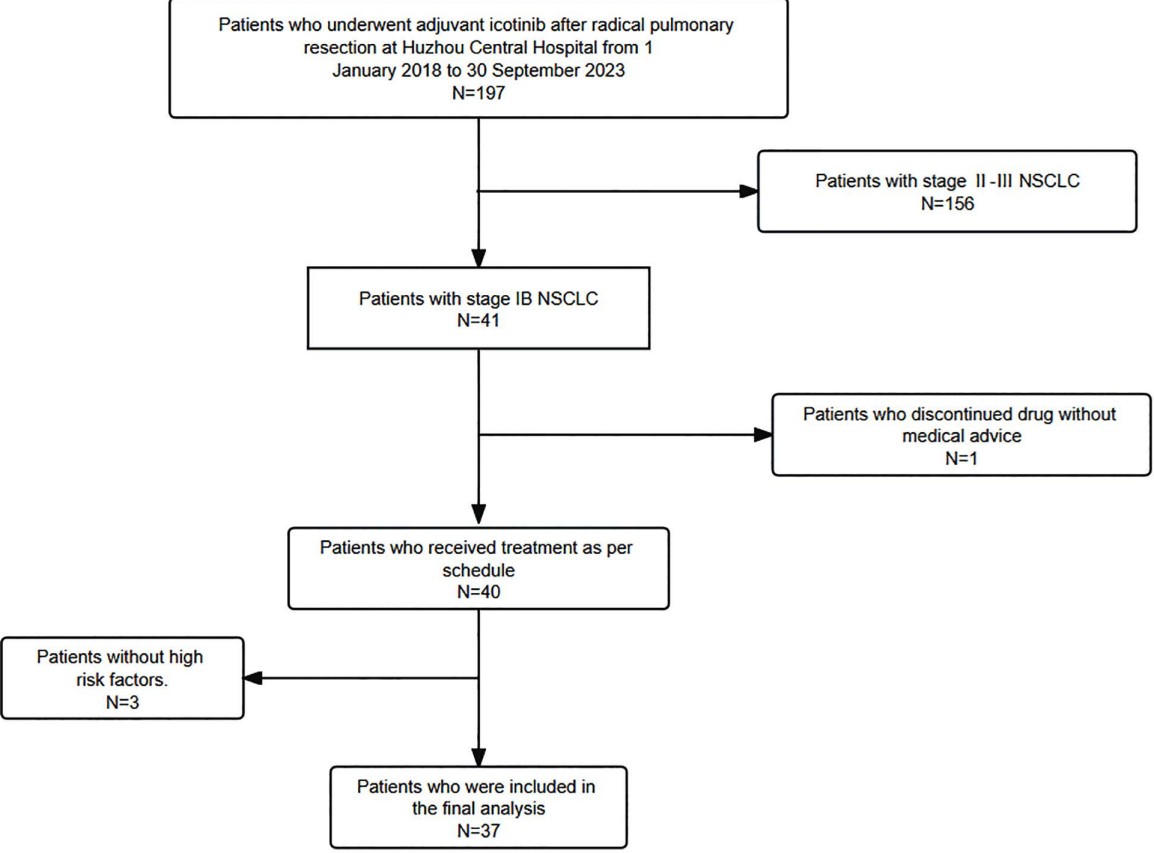

**Fig 1. Patient enrollment flowchart.** Based on the inclusion and exclusion criteria, 37 patients were finally enrolled in this study. The data cutoff date was March 1, 2025.

## Treatment and outcomes

The primary endpoint was the 3-year DFS, defined as the time from initiation of therapy to the first occurrence of disease recurrence or death from any cause. The secondary endpoints included CNS-DFS, OS, drug-related adverse events (AEs), and outcomes after disease relapse. OS was defined as the time from icotinib treatment to death from any cause. CNS-DFS was defined as the time from treatment to central nervous system metastasis. AEs were assessed according to the National Cancer Institute's Common Terminology Criteria for Adverse Events (CTCAE) version 5.0.

Icotinib was administered to all patients as first-line therapy one month after surgery at a dose of 125 mg thrice daily. The planned treatment duration was up to 36 months or until disease relapse or intolerable toxicity. Follow-up was conducted monthly via outpatient visits or phone calls to record adverse events (AEs). Chest computed tomography (CT), cranial magnetic resonance imaging (MRI), hepatic and adrenal ultrasounds, along with tumor markers detection were conducted every three months to assess disease status.

## Statistical analysis

Statistical analyses were performed with SPSS 26.0 software (IBM SPSS 26.0, SPSS Inc.). Descriptive statistics, including frequencies and percentages, were used for categorical variables. Continuous variables were expressed as

mean ± standard deviation (SD) or median and interquartile range (IQR). The DFS rate was calculated using the Kaplan-Meier method.

### Ethics statement

This study was reviewed and approved by the Research Ethics Committee of Huzhou Central Hospital, Affiliated Central Hospital of Huzhou University. Written informed consent was signed by all participants.

## Results

### Baseline characteristics

Based on the inclusion and exclusion criteria, 37 patients were finally enrolled in this study. The median age of the patients was 68 years (range 45–77 years). The cohort consisted of 16 (43.2%) male, 30 (81.1%) nonsmoker, and 37(100%) lung adenocarcinoma. Surgical procedures included 29 (78.4%) lobectomies, and 8 (21.6%) sublobectomies, meanwhile, systematic mediastinal lymph node dissection was performed in all patients. The median maximum tumor diameter was 26 mm (range 9–40 mm). All patients had at least one high-risk factor, such as LVI, STAS, VPI, or poor cellular differentiation. 17(45.9%) patients had EGFR exon 19-deletion mutations and 20(54.1%) patients had EGFR exon 21L858R mutations. No co-occurring genetic mutations were detected. The clinical characteristics were presented in Table 1 and Fig 2.

### Efficacy and safety assessment

The data cutoff date was March 1, 2025, and the median follow-up time was 31 months (range 18–74 months). The median treatment duration was 30 months (range 18–36 months). The 3-year DFS rate was 91.4% (95%CI：85%−99%), with no CNS metastasis detected at the time of data cutoff (Fig 3). Although the OS data were not yet mature, no death had occurred at data cutoff. Two patients experienced tumor recurrence at 28 and 30 months after initiating icotinib therapy, respectively. One patient was diagnosed with retroperitoneal lymph node metastasis, and the other with bone metastasis. Genetic testing of peripheral blood and bone biopsy tissue from both patients revealed EGFR-T790M mutations, indicating acquired resistance to icotinib and potential sensitivity to third-generation EGFR-TKIs. Following discussion by our Multi-Disciplinary Team (MDT), both patients received second-line therapy with osimertinib. No disease progression was observed duringr osimertinib treatment at the data cutoff. The characteristics of the patients with icotinib-resistant disease were presented in Table 2 and Fig 4. 29 (78.4%) patients experienced grade 1–2 AEs associated with icotinib, including rash in 15(51.7%), diarrhea in 10(34.5%), oral ulcer in 11(37.9%), paronychia in 5(17.2%), liver function damage 8(27.6%) and thrombocytopenia in 2(6.9%) patients. No grade 3 or higher AEs occurred (Fig 5).

## Discussion

According to the findings of the CORIN [8] trial, adjuvant icotinib can provide benefits to patients with Stage IB EGFR-mutated NSCLC. However, evidence regarding its efficacy in patients with high-risk factors is limited. To address this question, we conducted a real-world retrospective study to evaluate the effectiveness of adjuvant icotinib as a first-line therapy for high-risk stage IB EGFR-mutated NSCLC. The preliminary data showed that the 3-year DFS rate was 91.4% with favorable tolerance. No CNS metastasis or death were detected at the time of data cutoff.

The pathological stage is recognized as an independent prognostic factor for NSCLC [9]. Numerous studies [10–12] have demonstrated a correlation between pathological stage and survival outcomes following radical resection of NSCLC. Specifically, the 5-year survival rate for stage IB NSCLC is only 68% [13], and the survival rate is notably lower in patients presenting with high-risk factors, such as poorly differentiated tumors, including those with micropapillary or solid components [14], LVI [15,16], VPI [17,18] and STAS [19,20].

**Table 1. Clinical characteristics of patients (n = 37).**

| Characteristics | Median or n(%) |
|---|---|
| Age(year) | |
| Median | 68 |
| IQR[a] | 61.0-72.5 |
| Sex | |
| Man | 16 (43.2%) |
| Women | 21 (56.8%) |
| Smoking history | |
| None | 30 (81.1%) |
| Present | 7 (18.9%) |
| Tumor dimension | |
| Median | 2.6 |
| IQR[a] | 1.6-3.4 |
| Location of the tumor | |
| Right upper lobe | 12 (32.4%) |
| Right middle lobe | 7 (19.0%) |
| Right lower lobe | 3 (8.1%) |
| Left upper lobe | 9 (24.3%) |
| Left lower lobe | 6 (16.2%) |
| Operation | |
| Lobectomy | 29 (78.4%) |
| Sublobectomy | 8 (21.6%) |
| Pathological grading | |
| Moderately-differentiated | 27 (73.0%) |
| Poorly-differentiated | 10 (27.0%) |
| CTR[b] | |
| ≤0.5 | 6 (16.2%) |
| > 0.5 | 31 (83.8%) |
| Gene mutation | |
| 19-DEL | 17 (46.0%) |
| 21-L858R | 20 (54.0%) |
| VPI[c] | |
| None | 3 (8.1%) |
| Present | 34 (91.9%) |
| STAS[d] | |
| None | 33 (89.2%) |
| Present | 4 (10.8%) |
| LVI[e] | |
| None | 37 (100%) |
| Present | 0 (0%) |

[a]: IQR, Interquartile Range; [b]: CTR, consolidation tumor ratio; [c]: VPI, visceral pleural invasion; [d]: STAS, spread through air spaces; [e]: LVI, lymphovascular invasion.

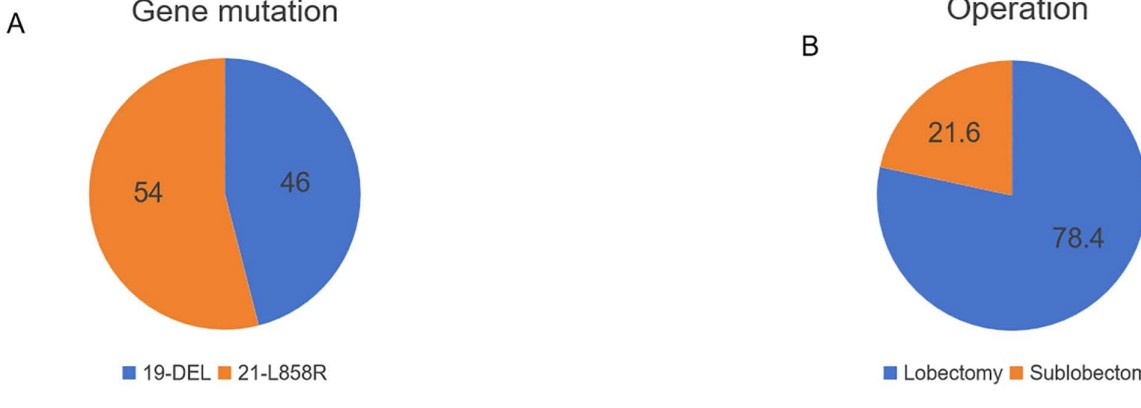

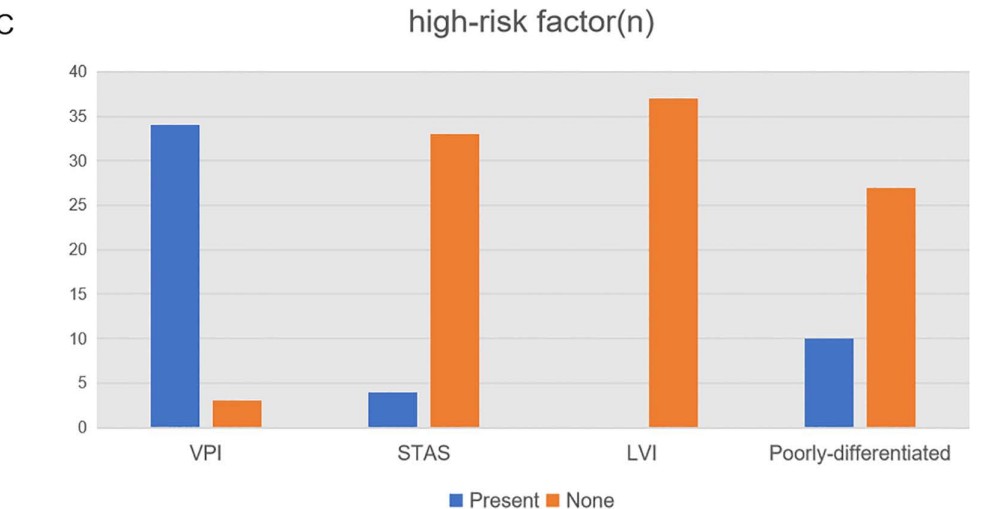

**Fig 2. Distribution of EGFR mutations, surgical approaches, and high-risk factors in this cohort.** (A) Frequency of EGFR mutation subtypes (19-DEL and 21-L858R), accounting for 46% and 54% respectively. (B) Proportion of patients undergoing lobectomy (78.4%) versus sublobectomy (21.6%). (C)The distribution of high-risk factors. VPI, visceral pleural invasion; STAS, spread through air spaces; LVI, lymphovascular invasion.

The application of postoperative adjuvant therapy for stage IB NSCLC remains controversial. Several previous studies [21,22] have suggested that the DFS and OS in patients with stage IB NSCLC may be improved with adjuvant chemo-therapy. Conversely, other research has failed to demonstrate a significant benefit from adjuvant chemotherapy for these patients [23,24]. While adjuvant chemotherapy is considered an optional strategy for improving the OS rates, the absolute improvement in 5-year survival with adjuvant cisplatin-based chemotherapy is only 5.4% [25]. Additionally, the incidence of severe AEs restricts its widespread application [26].

EGFR-TKIs have become the first-line therapy for advanced NSCLC with EGFR mutations [27–29]. Furthermore, adjuvant EGFR-TKIs have been recommended as an effective therapeutic approach for patients with early-stage EGFR-mutated NSCLC, including those with stage IB NSCLC [4]. Numerous retrospective studies have demonstrated the efficacy of postoperative adjuvant EGFR-TKIs in patients with EGFR-mutated stage IB NSCLC. Shen [30] et al. conducted a retrospective study on the effectiveness of adjuvant EGFR-TKIs in this patient population. Their findings revealed that the 3-year DFS rate was 98.3% in the EGFR-TKIs group, compared with 83.0% in the observation group

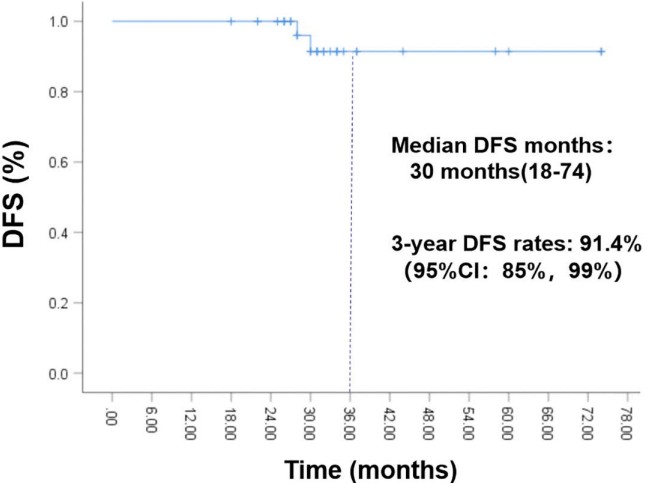

**Fig 3. Kaplan-Meier curve for DFS rate.** The 3-year DFS rate was 91.4%(95%CI: 85%−99%), with no CNS metastasis detected at the time of data cutoff. DFS, disease-free survival; CNS: central nervous system.

**Table 2. Clinical data of patients with disease recurrence.**

|  | Operation | Histological grade | Gene mutation | VPI[a] | STAS[b] | LVI[c] | DFS-[d](m) | OS[e](m) | Recurrence pattern | Second-line therapy |
|---|---|---|---|---|---|---|---|---|---|---|
| Case1 | Sublobectomy | Moderately-differentiated | 19-del | Present | None | None | 28 | 58 | retroperitoneal metastasis | osimertinib |
| Case2 | Lobectomy | Poorly-differentiated | 19-del | None | None | None | 30 | 47 | bone metastasis | osimertinib |

[a]: VPI, visceral pleural invasion; [b]: STAS, spread through air spaces; [c]: LVI, lymphovascular invasion; [d]: DFS disease-free survival; [e]: OS overall survival.

(HR: 0.10; 95% CI: 0.01–0.78; $p = 0.008$). Similarly, Jiang [31] et al. also reported a retrospective cohort study designed to assess the efficacy of adjuvant EGFR-TKIs in patients with stage IB EGFR-mutated NSCLC. The findings indicated that the 5-year DFS rate of the EGFR-TKIs group was significantly higher than that of the observation group (98.8% vs. 75.3%; P = 0.008). Recently, two prospective randomized controlled clinical trials have reported their outcomes concerning postoperative targeted therapy for early-stage EGFR-mutated NSCLC. The ADAURA [6,32,33] trial was specifically designed to assess the efficacy and safety of osimertinib in patients with completely resected stage IB-IIIA EGFR-mutated NSCLC. A subgroup analyses demonstrated that adjuvant osimertinib could significantly reduce the risk of relapse or death by 61% in the stage IB NSCLC. The CORIN [8] was a randomised, open-label, phase 2 trial specifically designed to compare adjuvant icotinib with observation in patients with stage IB EGFR-mutated NSCLC. The primary findings demonstrated that the 3-year DFS rate in the icotinib group was 96.1% (95% confidence interval [CI], 91.3–99.9), which was significantly higher than that in the observation group. However, none of the aforementioned studies conducted a stratified analysis of the outcomes of adjuvant EGFR-TKIs in high-risk stage IB NSCLC. In contrast, our study specifically enrolled patients diagnosed as stage IB NSCLC with high-risk factors, which distinguishes it from previous research. The preliminary analysis supported the efficacy of adjuvant icotinib with 91.4% for 3-year DFS rate. Two patients experienced tumor recurrence at 28 and 30 months after treatment initiation, respectively. No CNS metastases or deaths were observed at the data cut off.

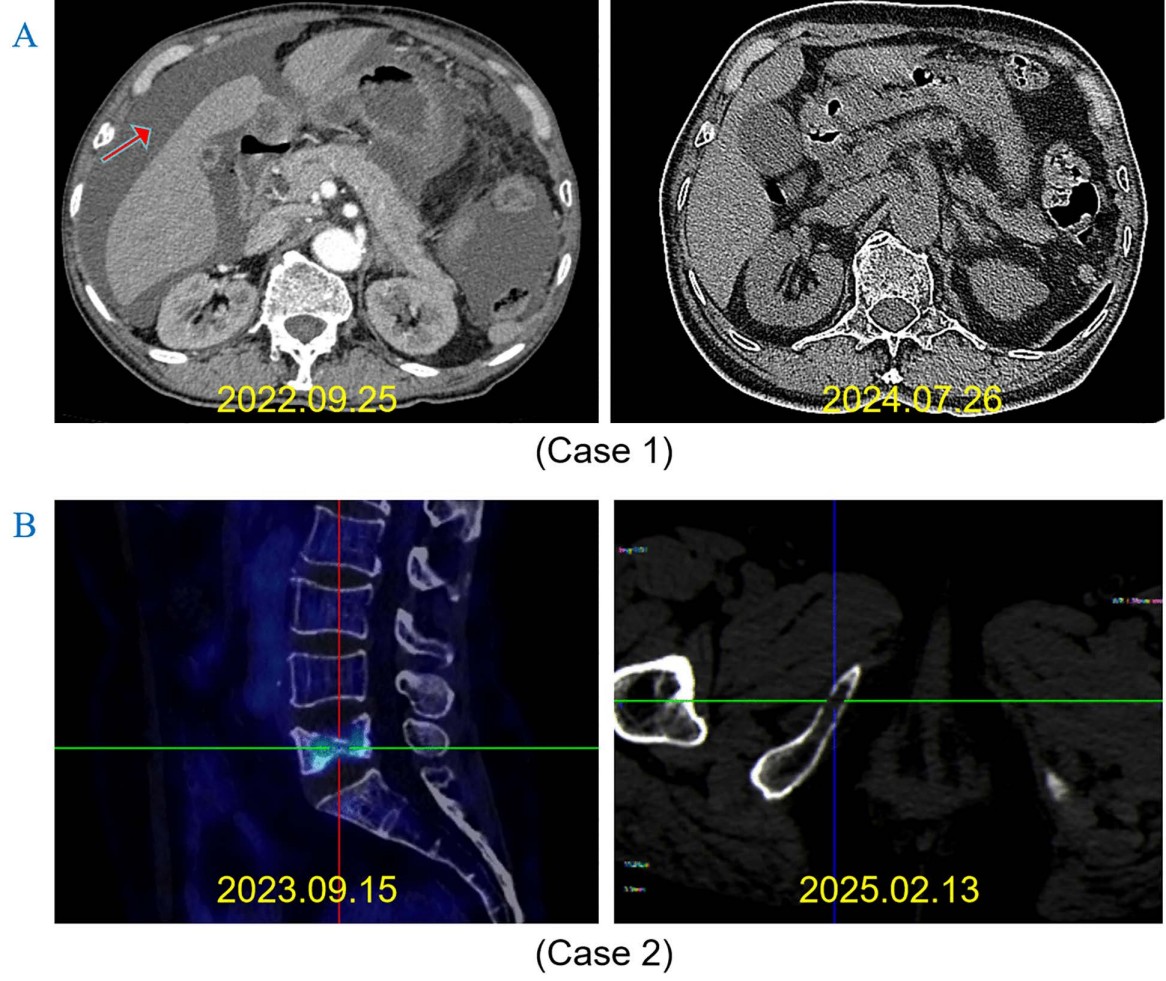

(Case 1)

(Case 2)

**Fig 4. Representative images of two patients successfully treated with osimertinib following acquired resistance to icotinib. (A)** Case 1: Retroperitoneal metastasis with malignant ascites. Contrast-enhanced abdominal CT scans before osimertinib initiation (left) and after 22 months of osimertinib treatment (right). **(B)** Case 2: Multiple bone metastases. PET-CT scan before osimertinib initiation (left) and bone scintigraphy after 17 months of osimertinib treatment (right). MRI: cranial magnetic resonance imaging; CT: Chest computed tomography; PET-CT: Positron Emission Tomography–Computed Tomography.

Nevertheless, there are still some limitations and biases in this study. Firstly, this is a single-center, single-arm study with the absence of a control arm, which precludes a direct assessment of the efficacy differences between chemotherapy and icotinib therapy. Secondly, a small sample size and the absence of a matched control cohort prevent us from drawing comparative conclusions about the efficacy of icotinib. Additionally, the long-term survival outcomes could not be assessed due to the relatively short follow-up period. Notwithstanding these limitations, our findings offer a strong rationale for further investigation. The next step is to validate and extend these results through a larger, multi-center case-control study with prolonged follow-up.

## Conclusions

This real-world study suggests that adjuvant icotinib is effective and safe in patients with high-risk stage IB EGFR-mutated NSCLC. Treatment-related AEs were manageable. Further follow-up is required to assess the long-term efficacy.

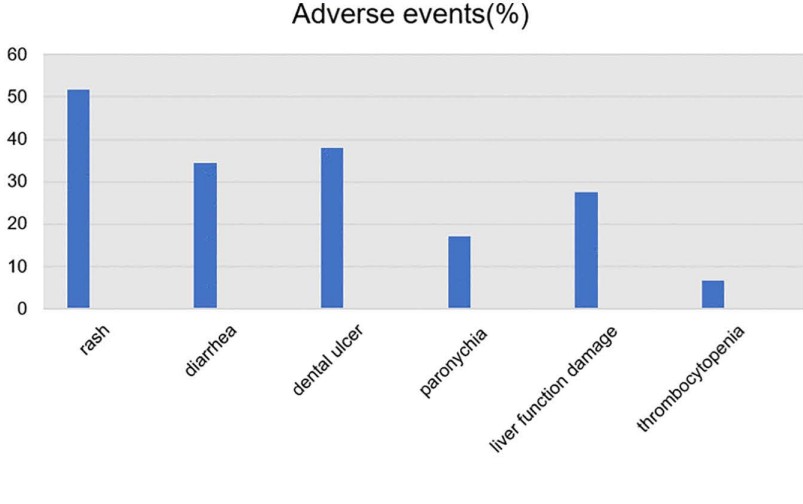

**Fig 5. Adverse events.** 29 (78.4%) patients experienced grade 1-2 AEs, including rash in 15(51.7%), diarrhea in 10(34.5%), oral ulcer in 11(37.9%), paronychia in 5(17.2%), liver function impairment in 8(27.6%) and thrombocytopenia in 2(6.9%) patients. No grade 3 or higher AEs occurred.

## Author contributions

**Data curation:** Mengzhi Cheng, Lili Jin.

**Methodology:** Mengzhi Cheng.

**Writing – original draft:** Mengzhi Cheng, Caihua Yu, Zhonghai Xie, Dong Li, Qinhua Gu.

**Writing – review & editing:** Jianbin Zhang, Qibin Shen.

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
