## [Decision Letter · Decision Letter 0]

1 Jul 2025

Dear Dr. Zhang,

Thank you for submitting your manuscript to PLOS ONE. After careful consideration, we feel that it has merit but does not fully meet PLOS ONE’s publication criteria as it currently stands. Therefore, we invite you to submit a revised version of the manuscript that addresses the points raised during the review process.

**ACADEMIC EDITOR: ** The reviewers have recommended publication, but also suggest minor revisions to your manuscript.  Therefore, I invite you to respond to the reviewers' comments and revise your manuscript.

We look forward to receiving your revised manuscript.

Kind regards,

Fumihiro Yamaguchi

Academic Editor

PLOS ONE

Journal Requirements:

2. Thank you for stating the following in your manuscript:

[This research was supported by the Joint Funds of the Zhejiang Provincial Natural Science Foundation of China under Grant No. LBY22H200006.]

[The author(s) received no specific funding for this work.]

Reviewers' comments:

Reviewer's Responses to Questions

**Comments to the Author**

1. Is the manuscript technically sound, and do the data support the conclusions?

Reviewer #1: Yes

Reviewer #2: Yes

2. Has the statistical analysis been performed appropriately and rigorously?

Reviewer #1: No

Reviewer #2: Yes

3. Have the authors made all data underlying the findings in their manuscript fully available?

Reviewer #1: Yes

Reviewer #2: Yes

4. Is the manuscript presented in an intelligible fashion and written in standard English?

Reviewer #1: Yes

Reviewer #2: Yes

Reviewer #1: The manuscript entitled “Adjuvant Icotinib in EGFR-mutated Stage IB Non-Small Cell Lung Cancer with High-Risk Factors: A Retrospective Case Series” by Cheng and colleagues presents a retrospective analysis evaluating the outcomes of adjuvant icotinib therapy in patients with resected stage IB EGFR-mutant NSCLC exhibiting high-risk pathological features. This study addresses a clinically important and timely question in thoracic oncology, especially in light of recent shifts toward targeted adjuvant therapies in early-stage disease. The authors report favorable disease-free survival and a tolerable safety profile, suggesting a potential benefit of prolonged icotinib treatment in this high-risk population. While the findings are promising, few issues require clarification to strengthen the scientific rigor and translational relevance of the study.

The first part of the result section can be visualized using demographic related figs.

The authors mentioned that chest computed tomography (CT), cranial magnetic resonance imaging (MRI), hepatic and adrenal ultrasounds, and tumor markers

detection were conducted every three months to evaluate the outcomes. Providing representative follow-up MRI or CT images at different time points (before, during and after treatment) with the Icotinib related to both responsive and resistant patient can complement current provided results.

Confidence interval and pvalue, log-rank or Cox regression not mentioned for DFS data

What method was used for EGFR mutation detection? blood or tissue?

The readability of paragraphs can be improved.

Was radiologic response assessed (e.g., RECIST criteria)? if yes please explain.

Reviewer #2: In this work, Cheng and colleagues explore the efficacy of adjuvant therapy with icotinib in patients with stage IB EGFR-mutated non-small cell lung cancer (NSCLC) who present high-risk factors, such as STAS, VPI, or poor differentiation. The inclusion and exclusion criteria are rigorous, enrolling only patients with high-risk stage IB NSCLC while omitting those with more advanced disease. The study offers a valuable real-world perspective, with detailed clinical follow-up and recurrence management (e.g., T790M mutation and switch to osimertinib).

The limitations of this study are clearly stated.

While preliminary, the findings are hypothesis-generating and may have clinical implications that warrant further investigation.

Minor Comments

• Please indicate the time interval between surgical resection and initiation of adjuvant therapy

• With the exception of patients who experienced a recurrence, did all others complete the 36-month treatment period? If not, please indicate whether any sub–grade 3 adverse events led to treatment discontinuation or modification, and specify the patient(s) involved

• Kindly revise the number of sublobectomies shown in Table 1, if inaccurate.

**Do you want your identity to be public for this peer review?** For information about this choice, including consent withdrawal, please see our Privacy Policy

Reviewer #1: No

Reviewer #2: No

---

## [Author Response · Author response to Decision Letter 1]

27 Jul 2025

Dear Editors and Reviewers,

We sincerely thank you and all reviewers for your valuable time and insightful feedback on our manuscript. We appreciate the opportunity to revise our work based on your constructive comments, which have significantly enhanced the manuscript and provided valuable inspiration for our future research endeavors. Please convey our gratitude to the reviewers. In response to the reviewers' comments, we have carefully revised the manuscript. We hope the revised version now meets the journal's standards. Below, we provide point-by-point responses to each comment and question.

To Reviewer #1

We are grateful for your critical evaluation and thoughtful suggestions, which have been instrumental in improving our manuscript. We have thoroughly addressed each point raised. Below are our detailed responses.

Comment 1: The first part of the result section can be visualized using demographic related figs.

Response: Thank you for highlighting this point. We have incorporated figures to visualize the demographic data presented in the first part of the results section, and the manuscript has been updated accordingly.

Comment 2: The authors mentioned that chest computed tomography (CT), cranial magnetic resonance imaging (MRI), hepatic and adrenal ultrasounds, and tumor markers detection were conducted every three months to evaluate the outcomes. Providing representative follow-up MRI or CT images at different time points (before, during and after treatment) for patients responsive and resistant to alectinib can complement current provided results.

Response: We appreciate this suggestion. As this study exclusively involved patients receiving adjuvant alectinib therapy following complete resection (stage IB), no post--treatment radiological assessments for response evaluation (as would be typical in neoadjuvant or metastatic settings) were performed. The imaging conducted every three months was for post-treatment surveillance of recurrence. Consequently, images depicting "before" or "during" treatment response are not available. However, as suggested, representative radiological images (CT/MRI) demonstrating recurrence in two patients have now been included as supplementary material.

Comment 3: Confidence interval and p-value, log-rank or Cox regression not mentioned for DFS data.

Response: We thank you for raising this methodological point. Given that this is a single-arm cohort study without a comparator group, formal statistical comparisons (e.g., log-rank test, Cox regression) for DFS were not applicable, and thus p-values and hazard ratios could not be calculated. The analysis presented is descriptive, reporting the observed DFS rates with confidence intervals and events.

Comment 4: What method was used for EGFR mutation detection? blood or tissue?

Response: EGFR mutation status was confirmed using a 10-gene next-generation sequencing (NGS) panel based on PCR-ARMS technology. This testing was performed on tumor tissue samples obtained at the time of surgical resection.

Comment 5: The readability of paragraphs can be improved.

Response: We thank you for this essential feedback. The language and readability throughout the manuscript have been carefully revised.

Comment 6: Was radiologic response assessed (e.g., RECIST criteria)? if yes please explain.

Response: We appreciate this query. Consistent with the study design focusing on adjuvant therapy after complete resection (stage IB disease), patients did not have measurable disease at the start of icotinib treatment. Therefore, formal radiologic response assessment using RECIST criteria was not applicable in this setting.

To Reviewer #2

We sincerely appreciate your critical review and valuable suggestions. We have carefully considered each comment and revised the manuscript accordingly. Our detailed responses are provided below:

Comment 1: Please indicate the time interval between surgical resection and initiation of adjuvant therapy.

Response: Thank you for this request for clarification. The adjuvant alectinib therapy was initiated one month after surgical resection.

Comment 2: With the exception of patients who experienced a recurrence, did all others complete the 36-month treatment period? If not, please indicate whether any sub–grade 3 adverse events led to treatment discontinuation or modification, and specify the patient(s) involved.

Response: We thank you for this important question. Not all patients who did not experience recurrence completed the full 36-month treatment period. The median duration of treatment was 30 months (range 18-36 months) at the time of data cutoff. However, no discontinuation of treatment or dose reduction was necessitated by adverse events of grade<3. Treatment modifications or discontinuations, where they occurred, were unrelated to toxicity below grade 3 severity.

Comment 3: Kindly revise the number of sublobectomies shown in Table 1, if inaccurate.

Response: Thank you for highlighting this point. The number of sublobectomies presented in Table 1 was indeed inaccurate and has been corrected accordingly in the revised manuscript and table 1.

---

## [Decision Letter · Decision Letter 1]

15 Sep 2025

Dear Dr. Zhang,

Thank you for submitting your manuscript to PLOS ONE. After careful consideration, we feel that it has merit but does not fully meet PLOS ONE’s publication criteria as it currently stands. Therefore, we invite you to submit a revised version of the manuscript that addresses the points raised during the review process.

The reviewers have recommended publication, but also suggest significant revisions to your manuscript.  Therefore, I invite you to respond to the reviewers' comments and revise your manuscript.

We look forward to receiving your revised manuscript.

Kind regards,

Fumihiro Yamaguchi

Academic Editor

PLOS ONE

Journal Requirements:

Reviewers' comments:

Reviewer's Responses to Questions

**Comments to the Author**

Reviewer #2: All comments have been addressed

Reviewer #3: (No Response)

2. Is the manuscript technically sound, and do the data support the conclusions?

Reviewer #2: (No Response)

Reviewer #3: Yes

3. Has the statistical analysis been performed appropriately and rigorously?

Reviewer #2: (No Response)

Reviewer #3: N/A

4. Have the authors made all data underlying the findings in their manuscript fully available?

Reviewer #2: (No Response)

Reviewer #3: Yes

5. Is the manuscript presented in an intelligible fashion and written in standard English?

Reviewer #2: (No Response)

Reviewer #3: No

Reviewer #2: (No Response)

Reviewer #3: There are still some grammatical errors but the manuscript is able to be read

The manuscript seeks to add to evidence about adjuvant TKI in resected IB NSCLC, in this case focussing on adverse pathological features. Below are my comments:

Line 58 Intro:

Update to 2024 or estimated 2025 figures

Remaining pertaining to intro and discussion:

Rather than focusing on just enlarging the sample, adding a cohort of matched controls with similar features could assist

The vast majority of patients had VPI as their adverse feature. It is not clear if the high disease control rate would extend to the other high risk features his should be addressed in the discussion as well of any relevant adverse features that were published in the sited series.

Would include stage II to IIIA, EGFR-mutated, non–small cell lung cancer adjuvant icotinib III trial (ICTAN, GASTO1002) presented at the 2024 American Society of Clinical Oncology (ASCO) Annual Meeting (Abstract 8004) to highlight that the optimal duration of use is unknown (even though these patients were stage II and III. This is relevant as the CORIN trial sited also has a more limited duration of therapy- 12 months. The ADAURA study, however, selected a 36 month duration of therapy which was the recommended duration of therapy your patients were receiving.

**Do you want your identity to be public for this peer review?** For information about this choice, including consent withdrawal, please see our Privacy Policy

Reviewer #2: No

Reviewer #3: No

---

## [Author Response · Author response to Decision Letter 2]

21 Sep 2025

Dear Editors and Reviewers,

We sincerely thank you and all reviewers for your valuable time and insightful feedback on our manuscript. We appreciate the opportunity to revise our work based on your constructive comments, which have significantly enhanced the manuscript and provided valuable inspiration for our future research endeavors.

Please convey our gratitude to the reviewers. In response to the reviewers' comments, we have carefully revised the manuscript. We hope the revised version now meets the journal's standards. Below, we provide point-by-point responses to each comment and question.

To Reviewer #3,

We are grateful for your critical evaluation and thoughtful suggestions, which have been instrumental in improving our manuscript. We have thoroughly addressed each point raised. Below are our detailed responses.

Comment 1: There are still some grammatical errors but the manuscript is able to be read.

Response: We thank you for this essential feedback. We have undertaken a comprehensive language polish of the manuscript, which involved not only correcting grammatical errors but also refining word choice, sentence structure, and overall clarity of expression to enhance readability. All modifications have been highlighted in red in the revised manuscript file. We hope the language quality of the manuscript is now significantly improved.

Comment 2: Line 58 Intro: Update to 2024 or estimated 2025 figures.

Response: We appreciate this suggestion. We have thoroughly searched for the GLOBOCAN 2024 or 2025 estimates. However, as of the preparation of this revision (September 2025), the most recent comprehensive global cancer incidence and mortality data available from the WHO/IARC is still the GLOBOCAN 2022 set, published in 2024. Therefore, we have maintained this citation and slightly rephrased the sentence to clarify the timeliness of the data source.

Comment 3: Rather than focusing on just enlarging the sample, adding a cohort of matched controls with similar features could assist.

Response: We thank the reviewer for this exceptionally constructive suggestion. We completely agree that the absence of a matched control cohort is a key limitation of our current retrospective study. In direct response to this comment, we have now explicitly acknowledged this limitation in the 'Discussion' section and have initiated plans to conduct a follow-up case-control study to address it in future work.

Comment 4: The vast majority of patients had VPI as their adverse feature. It is not clear if the high disease control rate would extend to the other high risk features his should be addressed in the discussion as well of any relevant adverse features that were published in the sited series.

Response: We thank the reviewer for raising this critical point. The reviewer is absolutely correct that the vast majority (91.9%) of our cohort presented with visceral pleural invasion (VPI). Consequently, our study is underpowered to perform a statistically meaningful subgroup analysis to determine whether the high disease-control rate extends equally to patients with other high-risk features, given the very low number of events within these small subgroups. Furthermore, upon reviewing the literature, we found limited evidence evaluating the prognostic influence of these specific high-risk features in the context of postoperative adjuvant targeted therapy. This gap further highlights the significance of our findings. Meanwhile, to address this limitation, we have cited relevant publications (e.g., [13-15, 18-19]) in the Discussion, affirming that features such as lymphovascular invasion, spread through air spaces, and poor differentiation are well-established independent adverse prognostic factors in early-stage NSCLC.

Comment 5: Would include stage II to IIIA, EGFR-mutated, non–small cell lung cancer adjuvant icotinib III trial (ICTAN, GASTO1002) presented at the 2024 American Society of Clinical Oncology (ASCO) Annual Meeting (Abstract 8004) to highlight that the optimal duration of use is unknown (even though these patients were stage II and III. This is relevant as the CORIN trial sited also has a more limited duration of therapy- 12 months. The ADAURA study, however, selected a 36 month duration of therapy which was the recommended duration of therapy your patients were receiving.

Response: We sincerely thank the reviewer for this critical and highly insightful comment, which has significantly improved the depth of our discussion. As suggested, we have included a discussion of the ICTAN trial findings in the Introduction section of our revised manuscript. Furthermore, we fully agree that the optimal duration of adjuvant TKI therapy remains a pivotal clinical question. To address this, we have expanded the Discussion section to explicitly acknowledge this uncertainty, citing the differing durations used in the CORIN (12 months), ICTAN (6-12 months), and ADAURA (36 months) trials. We clarify that the 36-month regimen used in our study was aligned with the ADAURA protocol and current clinical guidelines at the time of treatment. We also emphasize that future studies are needed to definitively establish the optimal treatment duration.

---

## [Decision Letter · Decision Letter 2]

6 Nov 2025

Adjuvant Icotinib in EGFR-mutated stage IB non-small cell lung cancer with high-risk factors: A retrospective case series

PONE-D-25-18906R2

Dear Dr. Zhang,

We’re pleased to inform you that your manuscript has been judged scientifically suitable for publication and will be formally accepted for publication once it meets all outstanding technical requirements.

Kind regards,

Fumihiro Yamaguchi

Academic Editor

PLOS ONE

Additional Editor Comments (optional):

Reviewers' comments:

Reviewer's Responses to Questions

**Comments to the Author**

Reviewer #3: All comments have been addressed

2. Is the manuscript technically sound, and do the data support the conclusions?

Reviewer #3: Yes

3. Has the statistical analysis been performed appropriately and rigorously?

Reviewer #3: Yes

4. Have the authors made all data underlying the findings in their manuscript fully available?

Reviewer #3: Yes

5. Is the manuscript presented in an intelligible fashion and written in standard English?

Reviewer #3: Yes

Reviewer #3: The authors have successfully expanded the discussion of relevant studies to support their conclusions. They also edited comments regarding the specific high risk features.

**Do you want your identity to be public for this peer review?** For information about this choice, including consent withdrawal, please see our Privacy Policy

Reviewer #3: No

---

## [Editor Report · Acceptance letter]

PONE-D-25-18906R2

PLOS ONE

Dear Dr. Zhang,

I'm pleased to inform you that your manuscript has been deemed suitable for publication in PLOS ONE. Congratulations! Your manuscript is now being handed over to our production team.

Kind regards,

on behalf of

Dr. Fumihiro Yamaguchi

Academic Editor

PLOS ONE